# Decision factors for the selection of AI-based decision support systems—The case of task delegation in prognostics

Kai Heinrich[1], Christian Janiesch[2]*, Oliver Krancher[3], Philip Stahmann[2], Jonas Wanner[4], Patrick Zschech[5]

1 Faculty of Economics and Management, Otto-von-Guericke-University Magdeburg, Magdeburg, Germany, 2 Department of Computer Science, TU Dortmund University, Dortmund, Germany, 3 Digital Business Innovation Section, IT University of Copenhagen, Copenhagen, Denmark, 4 Paxray GmbH, Mutlangen, Germany, 5 Faculty of Business and Economics, TUD Dresden University of Technology, Dresden, Germany

☯ All authors contributed equally to this work.

* christian.janiesch@tu-dortmund.de

## Abstract

Decision support systems (DSS) integrating artificial intelligence (AI) hold the potential to significantly enhance organizational decision-making performance and speed in areas such as prognostics in machine maintenance. A key issue for organizations aiming to leverage this potential is to select an appropriate AI-based DSS. In this paper, we develop a delegation perspective to identify decision factors and underlying AI system characteristics that affect the selection of AI-based DSS. Utilizing the analytical hierarchy process method, we derive decision weights for these characteristics and apply them to three archetypes of AI-based DSS designed for prognostics. Additionally, we explore how users' expertise levels impact their preferences for specific AI system characteristics. The results confirm that Performance is the most important decision factor, followed by Effort and Transparency. In line with these results, we find that the archetypes of prognostics systems using Direct Remaining Useful Life estimation and Similarity-based Matching best fit user preferences. Moreover, we find that novices and experts strongly prefer visual over structural explanations, while users with moderate expertise also value structural explanations to develop their skills further.

## 1. Introduction

Artificial intelligence (AI) has attracted substantial interest from both practitioners and scientists [1,2]. One driver of this interest is the increasing availability of large-scale data from widespread information technology (IT) and sensor technologies, which both enables and increasingly requires AI-based decision support [3]. By automating data analysis and pattern recognition, AI-based decision support systems (DSS) can reduce

**Data availability statement:** All relevant data are within the manuscript and its Supporting Information files.

**Funding:** This study was supported by the Federal Ministry of Education and Research (BMBF), Germany in the form of grants awarded to PZ (White-box AI: 01IS22080; AddIChron: 16SV8995). The specific roles of this author are articulated in the 'author contributions' section. The funder had no role in study design, data collection and analysis, decision to publish, or preparation of the manuscript.

**Competing interests:** The authors have declared that no competing interests exist.

the cognitive burden that large-scale data imposes on human decision-makers, improving thus the speed and quality of decisions—for instance, by predicting equipment failures in manufacturing and supporting timely maintenance actions [4].

A key activity in implementing an AI-based DSS is *selecting* an AI-based DSS that is suitable for the task. This is a complex decision: different DSS archetypes rely on different algorithms and offer varying trade-offs between predictive performance, explainability, implementation effort, and other factors [5]. For example, some systems, such as deep learning-based systems, can make highly accurate decisions, but they are typically inscrutable as decision-making is not transparent to the user [5–7].

Although existing research provides important perspectives on selecting AI systems for decision support, it addresses the selection problem in a limited way for four reasons:

First, the literature typically adopts an adoption or acceptance perspective, which does not fit well with the agentic and often inscrutable nature of AI. Models such as the technology acceptance model (TAM) [8] and the unified theory of acceptance and use of technology (UTAUT) [9] assume that users initiate system use voluntarily and retain full control over when and how to engage with a system. These models also assume that system functionality is transparent and knowable in advance. However, AI-based DSS challenge these assumptions: they often operate autonomously, make decisions without direct user initiation, and rely on inscrutable models that obscure how outputs are generated [10]. To better account for these characteristics, we follow Baird and Maruping (2021) to develop a delegation perspective that focuses specifically on the selection stage (cf. [11]).

Second, prior work has paid attention to a limited range of system characteristics influencing selection. While the trade-off between performance and transparency has received some attention [5,6], other AI system characteristics, such as the demands for computing resources and the training effort required to build an AI system, may also matter.

Third, there is little contextual research on how selection unfolds in specific application domains. One such domain is prognostics in machinery maintenance, where AI-based DSS support decisions with potentially high stakes, such as predicting equipment failures. Understanding selection in this context would provide practical insights by linking user preferences to archetypes of AI systems used in the field.

Finally and fourth, existing work has largely overlooked how user characteristics—especially expertise—shape system preferences. Cognitive psychology shows that novices and experts differ in how they process information and what forms of support they need [12,13], yet we know little about how differences in expertise affect preferences in selecting AI-based DSS.

Against this backdrop, our study takes a user-centered perspective to investigate how system characteristics and user expertise shape the selection of AI-based DSS in high-stakes decision-support contexts. While selection decisions in organizations often involve multiple actors, including managers and procurement specialists, users play a pivotal role: systems that do not align with their preferences are unlikely to be integrated successfully into everyday routines. To balance analytical focus with

practical relevance, we treat users as proxies for this range of organizational stakeholders, recognizing that their preferences can offer valuable insights into selection considerations. To ensure contextual realism, we situate our study in the high-stake domain of prognostics, where poor system choices can lead to significant financial losses or physical harm [14]. Specifically, we address the following research questions:

*RQ1: Which AI system characteristics are of importance in the selection of AI-based DSS for high-stake decisions?*

*RQ2: How do distinct archetypes of AI-based DSS for prognostics differ in these characteristics?*

*RQ3: How does user expertise influence the relative importance of AI system characteristics in the selection of AI-based DSS?*

To answer these questions, we develop a delegation perspective on AI-based DSS selection, drawing on the literature from other delegation contexts such as information system outsourcing. We then combine three methodological components: a multi-subject analytical hierarchy process (AHP) to elicit decision weights, an empirical evaluation of three implemented AI-based DSS archetypes, and a regression analysis to examine the moderating role of user expertise. This mixed-method approach balances the realism [15] of the detailed analysis of actual systems with the precision of AHP [16].

Our study makes four main contributions. First, it introduces a delegation-based perspective on AI-based DSS selection. Second, it offers empirical evidence on the relative importance of performance, effort, and transparency in system selection. Third, it links these findings to archetypes of AI-based DSS in the context of prognostics. Fourth, it reveals that the relationship between user expertise and preferences for transparency strategies is more nuanced than previously assumed.

The remainder of the paper is structured as follows: We begin with a review of relevant literature and the theoretical foundations for modeling DSS selection as a delegation problem. We then present our methodology, describe the use case, and operationalize the decision factors. Next, we report our empirical results, followed by a discussion of key implications. The paper concludes with a summary of contributions and directions for future research.

## 2. Background literature

### 2.1. Foundations of AI-based decision support systems

From an economic perspective, a decision involves selecting the best option under given circumstances [17]. In organizational contexts, decision quality depends on the availability and effective use of information. However, as information systems have proliferated, so too has the volume and complexity of information, often exceeding human cognitive limits [18,19]. DSS address this challenge by combining human judgment with computational analysis to support decision-making [20].

While there is a rich history of various types of DSS, such as rule-based DSS, recent advancements have led to AI-based DSS, which rely on machine learning (ML) algorithms trained on data rather than explicitly programmed rules [21,22]. These systems can uncover complex patterns autonomously and adapt to new data, thereby enhancing their effectiveness in dynamic environments. A notable development is the rise of deep learning, where neural network architectures automatically learn useful representations from high-dimensional inputs [22].

AI-based DSS excel in two important dimensions that drive business adoption. On the one hand, they can solve many problems with a *higher performance* (in various metrics) than traditional DSS. Examples range from process prediction [23] and credit scoring to load and sales forecasting [24]. On the other hand, they allow approaching complex problems and, thus, automating cognitive tasks that were not automatable before [25]. These abilities enable a more fine-grained management of decisions and task automation, which in turn enables novel monetization and cost-saving opportunities for businesses.

 

One driver of this performance is the shift from rule-based expert systems—where models are handcrafted using formal logic—to data-driven learning. Traditional systems struggle to codify tacit knowledge, limiting their applicability to complex tasks [26]. In contrast, ML-based systems learn from data and experience. Shallow ML algorithms require structured input and engineered features, while deep learning can bypass manual feature engineering by automatically learning representations from raw data. Engineering such systems involves specifying model architectures, curating training data, tuning hyperparameters, and training the model [22].

However, a major challenge lies in their lack of transparency. The non-linear structure of deep learning models often renders them inscrutable, making it difficult to understand which inputs drive their outputs [27]. These models are de facto "black boxes". As such, they can hinder user trust and acceptance, particularly in high-stakes contexts where decisions require justification. Explainability methods aim to address this by providing human-understandable representations of model logic [28,29]. Without such transparency, users may misunderstand, misuse, or reject system outputs [30,31]. Yet, explainability itself depends on the user's ability to interpret the explanations [32].

Explainability is often divided into *global* and *local* forms [33]. Global explanations offer visibility into an AI model's structural functionality, making the model's overall decision-making process transparent [34]. Local explanations make individual predictions transparent by providing visual, textual, or example-based explanations [28,35,36]. While global explanations typically support validating the AI model's processing by AI experts, local explanations rather present rationales for individual decisions to domain experts that are AI laypeople [37].

Other facets of AI-based DSS or intelligent systems in general discussed in extant literature are the increased capacity for agency [10], autonomy [38], or anthropomorphism [39] to solve tasks. These abilities are not inherent to the concept of ML but intertwined with their role and perception in respective scenarios and work processes as intelligent agents. For this reason, we did not consider these socio-technical facets attributed to emerging intelligent systems in our research.

## 2.2. work on decision factors for AI delegation and AI-based DSS selection

Recent research on AI has gained momentum across various themes, including learning in AI-based socio-technical systems, user acceptance, the distinct characteristics of AI, and augmentation and delegation. One stream investigates human-AI delegation and hybrid intelligence. Fügener et al. [40] examined human-AI delegation and found that hybrid intelligence outperforms both standalone human and AI-based DSS performance—but only when the AI makes the delegation decision, as humans often misjudge the capabilities of AI. They further show that AI advice to groups of humans does not yield the same performance gains, suggesting limited complementarities in such settings. Sturm et al. [41] and Grønsund and Aanestad [42] complement that there are several learning strategies for human-AI interaction. Van den Brock et al. [43] conducted an ethnographic identifying the dialectics and dynamics of hybrid practices involving human and AI knowledge.

A second stream focuses on selective AI use and acceptance. For instance, Vassilakopoulou et al. [44] examine human-AI interaction in chat services, while Ge et al. [45] suggest that successful workers are more inclined to adopt AI. Jussupow et al. [46] examine the metacognition of physicians while they deal with AI advice. However, these works typically assume a specific AI system is already in place and do not address system selection.

Other studies theorize the socio-technical distinctiveness of AI. Berente et al. [47] contemplate the frontiers of managing AI, specifically depicting facets like autonomy and inscrutability to distinguish AI systems from classical information systems. Schuetz and Venkatesh [48] argue that AI systems are distinct from other software in their ability to interact with, learn from, and adapt to the environment without much human intervention. Furthermore, from an agency perspective, AI systems have an increased presence in terms of automaticity, rationality, and endorsement when compared to traditional information systems [49].

Some frameworks address human-AI interaction more broadly. Jain et al. [50] introduce an intelligence augmentation framework categorizing different use scenarios. Others, including Baird and Maruping [10], Lyytinen et al. [51], and

Schuetz and Venkatesh [48], propose delegation frameworks that contextualize agentic information systems and cognitive computing challenges, in line with Ågerfalk's [52] call to anchor AI in theory of the Information Systems discipline. However, none of these frameworks offer concrete guidance on selecting among AI-based DSS.

The study most closely related to model choice is by Asatiani et al.'s [6], who examine the use of AI at the Danish Business Authority. They describe a trade-off between accuracy and transparency in AI selection and outline governance practices such as periodic audits to ensure responsible use of less transparent systems.

Finally, user expertise has emerged as a relevant factor in AI use studies. Research shows that expertise plays a critical role in ill-structured but structurable tasks—such as selecting among AI-based DSS—because experts process complex information more effectively, detect meaningful patterns, and build robust mental models [53]. This is due to the fact that experts can perceive large meaningful patterns in their domain and can build mental models of the problem, they are fast and make little errors and have strong self-monitoring skills, they have superior problem-related short-term and long-term memory, and they are able to see a problem at a deeper and more principled level than novices. As a caveat, research has also shown that experts excel mainly in their own domain [54]. Taken together, there is ample evidence that the user's expertise is a key factor shaping user's attitudes and their use of AI systems.

In summary, while existing work identifies the performance-transparency trade-off, the role of expertise, and the need for monitoring, it pays little attention to the concrete selection of AI-based DSS—especially in relation to different system archetypes and user characteristics.

## 3. Conceptual foundations

### 3.1. AI-based DSS selection as part of a delegation problem

Drawing on Baird and Maruping [10] as well as Dibbern et al. [55], we conceptualize the selection of AI-based DSS as a stage in a delegation process. Similar to outsourcing decisions, selecting an AI-based DSS involves choosing a (technical) actor to perform a task previously handled by a human. For example, when machine maintainers rely on AI recommendations, they delegate data analysis and prioritization to the system. Unlike traditional IT use cases, such delegation introduces concerns about system autonomy and inscrutability [38,49], requiring ongoing monitoring.

Viewing AI-based DSS selection as a delegation problem allows us to draw on established theories—agency theory [56,57], transaction cost economics [58], and the resource-based view [59]—originally developed for inter-organizational outsourcing but also applied in other delegation contexts (e.g., employment, consulting, IT projects) [60]. There are important commonalities with delegation to AI-based DSS. Analyzing these commonalities aids in theoretically deriving the characteristics of AI-based DSS and users that influence selection.

A first commonality is the *trade-off between delegation benefits and associated risks*. As with outsourcing, where tasks are delegated for efficiency or quality gains [55,58], adopting AI-based DSS can improve predictive performance—but often requires investment in implementation and training [61]. This implies that both *expected performance* (e.g., accuracy, speed) and expected *effort* (e.g., implementation time, skill requirements) influence selection.

Second, *limited behavioral observability* plays a role in both settings. Agency theory highlights the need for principals to monitor agents when behavior is not directly observable [57,62]. Similarly, AI-based DSS with opaque models require users to invest effort in understanding or verifying decisions [5,63]. *Transparency* thus becomes a critical selection factor.

Third, the challenge of adverse selection applies to both traditional delegation and AI-based DSS selection. In economic theory, principals face *uncertainty about an agent's true abilities*, such as their skills at the time of hiring, and therefore prefer agents whose capabilities are easier to verify [57]. Similarly, users may be uncertain whether an AI system that performs well on one dataset will generalize to new data [64]. This uncertainty makes verification crucial. However, verifying an AI system's capabilities often requires significant effort—such as inspecting recommendations or retraining models—which users may seek to minimize. As a result, users may prefer AI-based DSS that are easier to verify through higher transparency [5], highlighting a trade-off between *effort* and *transparency* in selection decisions.

Fourth, *human information processing* plays a crucial role in the delegation decision. Agency theory indicates that the principal's expertise allows them to interpret information about the agent's behavior, crucial for exercising control [62,65,66]. It has also been shown that principals benefit from different types of information depending on their knowledge in the task domain [67]. Similarly, the user's expertise may influence the types of information they find helpful in monitoring the AI-based DSS [68]. Thus, the user's expertise may affect AI-based DSS selection.

### 3.2. Decision factors and AI system characteristics for delegation

Based on these commonalities, we identify four key factors: three relate to AI system characteristics—Performance, Effort, and Transparency—and one relates to user characteristics—Human Information Processing. Table 1 summarizes these factors and their links to delegation theory.

**3.2.1. Performance.** Key potential benefits of the use of an AI-based DSS lie in making more accurate decisions in a shorter time [69]. We capture these benefits in two indicators: Predictive Accuracy and Inference Time (e.g., [70]).

*Predictive Accuracy* refers to the system's ability to generate correct predictions based on input data. Higher accuracy improves the likelihood that the AI-based DSS will provide meaningful support in decision-making tasks, particularly in complex, data-intensive domains (e.g., [70]).

*Inference Time* denotes the time a trained model requires to process new input data and produce an output. Faster inference times enable users to respond more quickly, which is especially critical in time-sensitive decision contexts.

Ceteris paribus, decision-makers will prefer AI-based DSS with higher Predictive Accuracy and shorter Inference Time. both Predictive Accuracy and Inference Time may evolve over time as AI systems learn and adapt based on new data and parameter tuning (e.g., [69,71]).

**3.2.2. Effort.** The potentially heightened Performance from using an AI-based DSS comes with the Effort needed to implement the system [72]. High effort makes it more difficult to verify the capabilities of the AI-based DSS [73]. We distinguish three effort-related characteristics:

*Required Skill Level* captures the degree of expertise needed to understand and work with a particular AI-based DSS. Higher required skill levels imply greater demands not only for data scientists but also for engineers and domain experts involved in system implementation and use.

**Table 1. Factors Affecting the Selection of AI-based DSS.**

| | Factor | Characteristic | Definition | Link to Delegation Perspective |
|---|---|---|---|---|
| AI System Characteristics | Performance | Predictive Accuracy | The system's prognostic power | Increased Predictive Accuracy and reduced Inference Time are potential delegation benefits |
| | | Inference Time | The time the system requires to produce an output. | |
| | Effort | Implementation Time | The time required to build the system | Implementation Time, Training Time, and Required Skill Level make it more effortful to verify capabilities |
| | | Training Time | The time required to train a model with training data | |
| | | Required Skill Level | The effort needed to understand and implement the system | |
| | Transparency | Structural Explanation | The system's capability of explaining its structural functionality by providing transparency about the system's overall decision logic, its components such as parameters, and its algorithm | Structural Explanation and Visual Explanation can increase behavior observability and, thus, reduces monitoring efforts; they can also make it easier to verify capabilities |
| | | Visual Explanation | The system's capability of providing instance-based graphical explanations for individual decisions | |
| User Characteristics | Human Information Processing | Expertise | The user's knowledge and ability to act competently in the realm of the task | Expertise affects human information-processing and, thus, which information is suitable for monitoring and verification |

*Implementation Time* refers to the time and effort required to build and configure the model, including the selection of parameters and algorithm components.

*Training Time* is the time needed to train the system using appropriate datasets, which may be substantial for complex models.

These three elements jointly determine the practical difficulty of setting up a working AI-based DSS. Greater effort not only increases implementation costs but also makes it harder to verify a system's capabilities, potentially discouraging adoption [73].

**3.3.3. Transparency.** Transparency is a critical factor in AI-based DSS selection, as users must be able to monitor and assess the system's behavior. Opaque black-box models, such as those based on deep learning, obscure the reasoning behind specific outputs and make it harder for users to validate system performance [27]. Regarding AI-based DSS, inscrutability of the decision logic of black-box models makes it more difficult for the user to validate whether the system operates as it should [74]. Much like the principal is uncertain of the true qualities of an agent in other delegation settings such as outsourcing (adverse selection), the user of a black-box AI-based DSS is uncertain about the AI-based DSS' true qualities, including the question of whether the algorithm will perform well on decision making tasks beyond the training data set. Although users may respond to this by creating transparency through monitoring (e.g., checking all decisions made by the AI-based DSS), high monitoring costs may erode the benefits of using AI-based DSS. A more efficient system is to improve observability of the AI-based DSS.

Designers can increase the transparency of an AI-based DSS through measures of algorithmic self-signaling by the AI-based DSS. This call for improved observability of the decision logic of an AI-based DSS is in line with the explainable AI (XAI) community's demand for explainable models while maintaining a high degree of model predictability [75]. In this case, explainability is about the understanding of the reasoning behind a decision by a human observer [76]. Thus, the degree of model explainability remains dependent on the respective end user. As argued above, there is a distinction between two types of explanations: global explanation and local explanation [77]. The former offers a *Structural Explanation*. It is about offering functionality-based transparency into the overall decision logic of an AI-based DSS by providing information about its components such as parameters, or the learning algorithm and its explanatory value [78]. The latter is about providing instance-based explainability of individual predictions of an AI-based DSS through textual explanations, graphical presentations, or examples [35]. It aims at an appropriate type of local explanation, most commonly a *Visual Explanation* for the intended end users [79,80].

**3.3.4. Human information processing.** Research on delegation problems, such as information system control and sourcing, indicates that the principal's expertise influences their interpretation and action on different types of information [62,65–67]. Cognitive psychology research supports this, showing significant differences in cognitive processes between novices and experts in a domain, who differ in the years of engagement with a defined topic [12,81,82]. Consequently, information that benefits experts may be inconsequential or counterproductive for novices [13,83]. Thus, we propose that the user's expertise affects their ability to interpret information for monitoring AI-based DSS and verifying capabilities, influencing their preference for transparency strategies, viz., Structural vs. Visual Explanation.

## 4. Study design

### 4.1. Overview of research methods

Our study combines three methodological components to address the research questions: (1) an AHP to elicit decision weights (RQ1), (2) the empirical implementation of three AI-based DSS archetypes in a high-stakes prognostics case (RQ2), and (3) a regression analysis to examine the moderating role of user expertise on preferences (RQ3). Fig 1 provides an overview of the research methodology.

To address RQ1—identifying which decision factors influence the selection of AI-based DSS—we structured the selection decision as a hierarchical model. Drawing on agency theory, transaction cost economics, and the resource-based

view, we defined key decision factors and associated system characteristics (e.g., Transparency and its sub-characteristics Structural and Visual Explanation). Fig 2 illustrates the resulting hierarchy, which served as the basis for the AHP analysis [16].

To address RQ2—understanding users' relative preferences for different AI-based DSS—we developed three real-world DSS archetypes for a predictive maintenance scenario (see Section 4.2). We operationalized the AI system characteristics (Section 4.3) and collected users' pairwise comparisons of both decision factors and system characteristics using Saaty's scale. These comparisons produced individual weights, which we aggregated using the method by Dijkstra [84] applying these weights to the measured characteristics of each DSS archetype allowed us to derive user-based rankings of the alternatives (Section 4.4).

To address RQ3—examining the influence of user expertise on delegation preferences—we compared preferences for system characteristics across user segments with different expertise levels. We conducted probit regression analyses to assess the statistical significance of these differences while controlling for demographic.

## 4.2. Use case of high-stake condition-based maintenance

To provide contextual realism and ensure comparability of AI-based DSS alternatives, we embedded our study in the high-stakes domain of predictive maintenance. This domain is characterized by the need to anticipate equipment failure before it occurs—decisions that can involve severe consequences, including financial loss or bodily harm [14]. For example, downtime may cost up to $250,000 per hour [85] and failure of critical systems like turbine engines can have catastrophic consequences [4]. Maintenance decisions often rely on prognostic tools that estimate the remaining useful life (RUL) of components such as aircraft turbines. These decisions involve considerable uncertainty and require

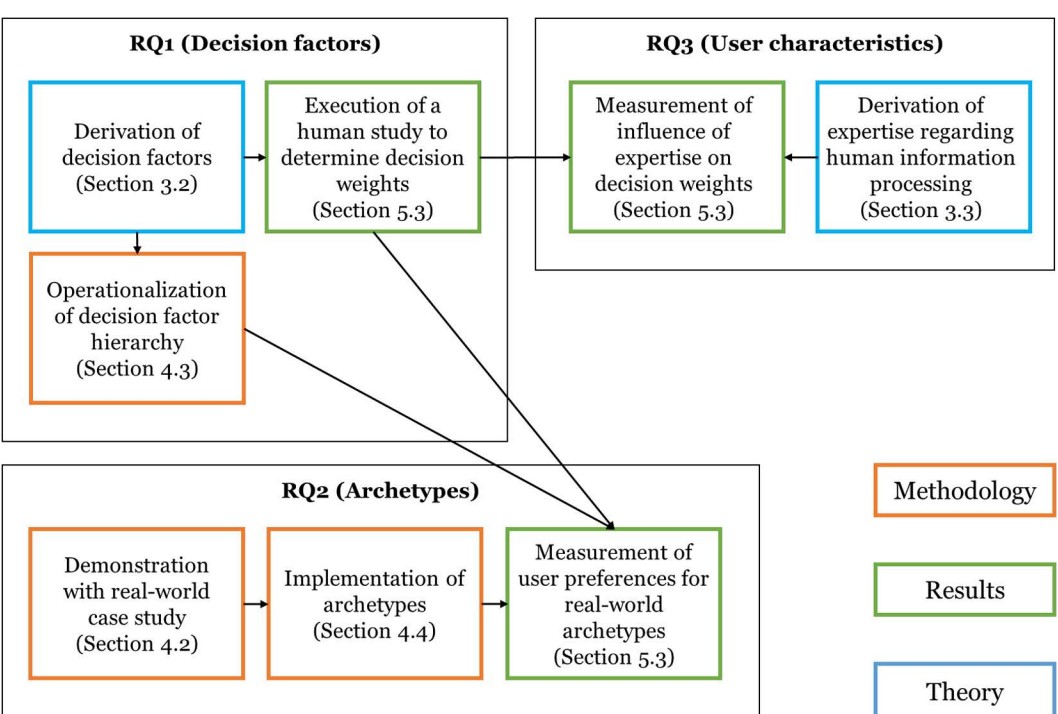

**Fig 1. Research Methodology.**

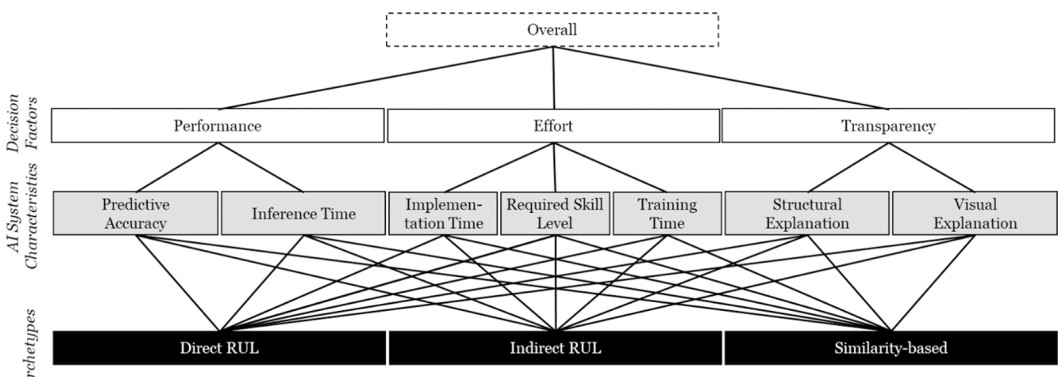

**Fig 2. Hierarchical Model of the Decision Process for the AHP.**

balancing performance, effort, and transparency—making this domain highly suitable for studying user preferences toward AI-based DSS.

Our implementation builds on the *Commercial Modular Aero-Propulsion System Simulation (C-MAPSS)* dataset provided by NASA's Prognostics Center of Excellence. The C-MAPSS dataset simulates sensor readings and degradation patterns from turbofan engines under varying operational conditions and fault modes. It is a recognized benchmark in the prognostics community and has been used in numerous studies to evaluate predictive models for RUL estimation [86]. Its public availability and high quality make it well suited for replicable, scenario-based decision experiments.

### 4.3. AI-based DSS archetypes

For the identification of AI-based DSS archetypes and best practices implementing these archetypes, we looked into the review papers from Ramasso and Saxena [87] and Zschech et al. [88]. On this basis, we identified three prevailing archetypes of AI-based prognostic systems: Direct RUL, Indirect RUL, and Similarity-based Matching. Each archetype is designed as an AI-based DSS, consistent with our delegation perspective, and differs in terms of performance, effort, and transparency characteristics.

The *Direct RUL* archetype uses a deep learning model—specifically, a long short-term memory (LSTM) network—to directly predict RUL from raw sensor data. This approach is known for its high predictive accuracy and is widely used in the prognostics literature [88]. However, it requires substantial implementation effort and offers limited transparency to end-users.

The *Indirect RUL* archetype follows a two-step approach. First, a classifier predicts discrete health states based on sensor inputs. Second, a rule-based mapping translates these states into RUL estimates. This method reduces complexity by separating state classification from RUL estimation and allows for modular development. Although it is somewhat easier to implement and verify, it tends to lag behind direct prediction in terms of accuracy.

The *Similarity-based Matching* archetype implements a *k*-nearest neighbor approach to identify degradation trajectories similar to the current sensor readings. This method emphasizes intuitive explanation through comparison, making it more transparent for users unfamiliar with black-box models. While this archetype may not always achieve top-tier accuracy, it provides high-quality visual explanations and requires relatively low implementation effort—factors relevant for delegation and trust.

All models were implemented using standardized Python libraries and evaluated on the same subset of the C-MAPSS dataset to ensure comparability. See S1 Appendix Part B for technical details on the implementations.

## 4.4. Operationalization of decision factors

To investigate which system characteristics matter to users when selecting AI-based DSS (RQ1, RQ2), we operationalized the three theoretical decision factors—Performance, Effort, and Transparency—along with their associated system characteristics. These factors, derived from the delegation perspective and detailed in Section 3.2, were made measurable to enable empirical comparison of system alternatives and user preferences.

*Performance* was measured using two indicators: predictive accuracy and inference time. Predictive accuracy refers to the prognostic power of a model. We employed two measures commonly used in RUL prediction challenges: the PHM08 score and root mean squared error (RMSE) [88]. The PHM08 score emphasizes early failure prediction by penalizing late predictions more heavily, with an exponential penalty that increases with the magnitude of the error [86]. This scoring function reflects a risk-averse perspective that prefers conservative, early warnings. However, it does not consider the prognostic horizon—the time between the prediction and the actual failure [89]. RMSE was used in parallel to counterbalance this bias, as it penalizes both early and late deviations symmetrically and helps identify models that systematically underestimate the RUL. Together, the two measures allow for a more nuanced assessment of accuracy. Inference time was computed as the average time (in milliseconds) a trained model required to generate an output after receiving input. All systems were tested under the same technical conditions to ensure comparability.

*Effort* was captured through three characteristics: required skill level, implementation time, and training time. Required skill level was qualitatively evaluated using predefined criteria reflecting the technical knowledge necessary to implement and interact with each system. Implementation time referred to the time needed to configure each model and integrate it into a functioning pipeline. Training time reflected the time required to train the model on the provided dataset. Together, these characteristics represent the practical resource demands of deploying each DSS archetype.

*Transparency* was assessed through the presence and accessibility of structural and visual explanations. Structural explanation was evaluated based on the availability of global insights into each model's decision logic, such as interpretability of parameters and model architecture. Visual explanation was measured based on user evaluations of each system's outputs. As part of the data collection process, participants reviewed outputs from all three AI-based DSS archetypes and ranked their clarity and usefulness in helping them understand individual predictions (see Section 4.5).

To address RQ3, we measured human information processing in terms of user expertise. Participants reported their experience in industrial maintenance using a 5-point Likert scale ranging from "<2 years" to ">15 years." This self-reported domain expertise served as a proxy for their ability to interpret and act upon complex information when evaluating AI-based DSS.

All operationalized values were normalized to facilitate integration with the AHP-based decision weights. S1 Appendix Part A provides further details on measurement formulas.

## 4.5. Study procedure

To elicit user preferences and evaluate AI-based DSS selection, we conducted a four-step experimental study combining scenario-based tasks with the AHP method (RQ2, RQ3). The procedure is illustrated in Fig 1.

**Step 1: Demographics and Expertise Assessment.** Participants completed a questionnaire capturing demographic data (e.g., age, gender, industry), general AI attitudes, and self-rated domain and AI expertise. This information was used to investigate the moderating role of expertise (RQ3).

**Step 2: Scenario Briefing and Explanation of Characteristics.** Participants were introduced to the predictive maintenance case and received explanations of each system characteristic. Examples of inference time, visual explanation outputs, and predictive accuracy were provided to enable informed comparisons.

**Step 3: Visual Ranking Task.** To establish subjective preferences for visual explanations, participants were shown the outputs of all three AI-based DSS archetypes on sample cases and asked to rank them. These rankings informed our

transparency operationalization. Fig 3 shows an example for the visual presentation of results of the AI-based DSS. The *y*-axis represents the health status estimated by the AI system. The *x*-axis indicates the amount of flights of the turbine. Red crosses represent the failures known to the system from the training data, whereas the large orange cross is the calculated failure of the turbine, which implies its RUL.

**Step 4: AHP Pairwise Comparisons.** Finally, participants completed a series of pairwise comparisons between decision factors and between system characteristics within each factor, using Saaty's (2008) 1–9 scale. This produced individual decision weights at both levels of the hierarchy. To reduce fatigue and ensure data quality, comparisons were structured and kept within recommended limits for consistency Dijkstra [84].

### 4.6. Sample and recruitment

To collect preference data from a relevant population, we recruited participants through the Prolific platform. Participants were prescreened to ensure prior experience in industrial maintenance or engineering domains. To enhance validity, we applied an additional filter requiring participants to have completed at least 50 prior tasks on Prolific with an approval rate above 95%.

Before accessing the main part of the study, participants were presented with a brief screening questionnaire to verify relevant experience. Participants who failed to meet the expertise requirement or completed the study in under six minutes were excluded. A total of 105 participants began the study, of whom 83 completed it in full and passed all attention and timing checks.

Participants were compensated in line with Prolific's recommended hourly rate of approx. $14/hr. The sample consisted of participants from various English-speaking countries, with diverse age, gender, and professional backgrounds. Descriptive statistics for the final sample are reported in Table 2 (see Section 5.1).

For anonymous Internet-based surveys, it is usually appropriate to use implied informed consent. Subject consent is implied by submitting the completed survey. The subjects could terminate the survey at any time. Consequently and in conformity with Information Systems research practice, approval by an ethics committee was not sought. Data collection was from 24/04/2020 to 25/04/2020.

### 4.7. Data aggregation and analysis

To compute overall user preferences for each AI-based DSS archetype, we combined the individual decision weights from the AHP with the normalized system performance scores. This yielded an aggregated preference score for each

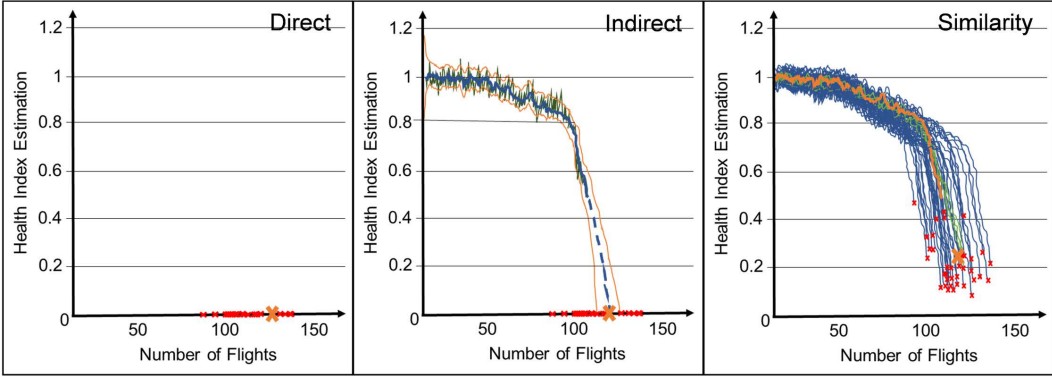

**Fig 3. Visual Explanation of Each Prognostic Approach.**

**Table 2. Decision Weights of Decision Factors and Characteristics.**

| Decision Factor | AI System Characteristic | Decision Weight ($n=120$) | |
|---|---|---|---|
| Performance | Predictive Accuracy | **0.901** | **0.608** |
| | Inference Time | 0.099 | |
| Effort | Implementation Time | 0.215 | 0.210 |
| | Required Skill Level | 0.548 | |
| | Training Time | 0.236 | |
| Transparency | Structural Explanation | 0.241 | 0.181 |
| | Visual Explanation | 0.759 | |

participant and system alternative. AHP weights were calculated using the standard eigenvalue method [16], and aggregation followed the procedure described by Dijkstra (2013).

Additionally, we implemented a consistency check by calculating the consistency index for each participant's AHP responses. Following Saaty (2008), we flagged any cases with an inconsistency rate above 10%. No such inconsistencies were found, which can be attributed to the simplicity of our pairwise comparisons—five comparisons at the decision factor level and three at the characteristic level.

To analyze how user expertise affects preferences (RQ3), we ran probit regressions using the binary preference for each AI-based DSS archetype as the dependent variable. Independent variables included user expertise (measured on a five-point Likert scale) and controls for age, gender, industry background, and risk preference.

## 5. Results

### 5.1. Descriptive statistics

The final sample consists of $n=120$ participants. These respondents were selected from an initial group of 165 individuals. 45 participants were excluded for not meeting the minimum completion time of 5 minutes, which we set to ensure proper engagement with the material. No additional exclusions were necessary due to patterned or careless responses. Participants were predominantly based in Europe (66%) and North America (33%), with 1% from other regions. Gender distribution was 63% male, 35% female, and 2% identifying as diverse. Age groups were broadly distributed: 23% were aged 20–30, 33% were 31–40, 25% were 41–50, 18% were 51–60, and 3% were older than 60. The average survey duration was 9.17 minutes (median = 6.17, SD = 4.7). To assess domain-specific expertise, participants reported their experience in industrial maintenance on a 5-point Likert scale. The sample included 19% with 2–5 years, 22% with 6–10 years, 3% with 11–15 years, and 12% with over 15 years of relevant experience.

### 5.2. Importance of decision factors

To address RQ1, we calculated individual and aggregated decision weights for the three main decision factors—Performance, Effort, and Transparency—using the AHP method. Performance emerged as the dominant factor, with a mean decision weight of 0.61, followed by Effort (0.21) and Transparency (0.18). These weights reflect the relative importance users assigned when selecting among AI-based DSS options in a high-stakes decision context.

Fig 4 presents the distribution of responses for the pairwise comparisons between the decision factors. Most respondents considered Performance more important than Transparency (Fig 4 (a)) and Effort (Fig 4 (b)). The comparison between Transparency and Effort showed a more balanced distribution, with a slight preference for Transparency (Fig 4 (c)).

Table 2 details the decision weights for each AI system characteristic nested within the three decision factors. We observed the highest within-factor polarization for Performance, where Predictive Accuracy (0.901) was strongly preferred

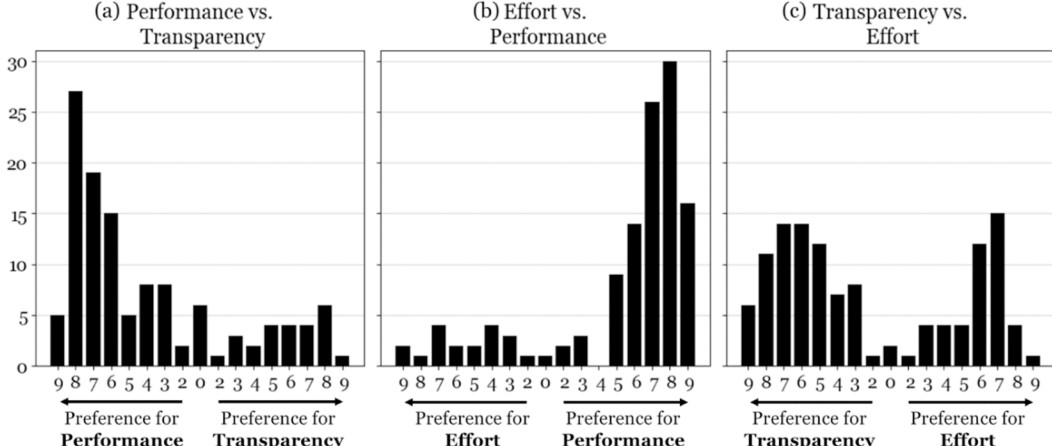

**Fig 4. Answer Distributions in the Pairwise Comparisons of Decision Factors.**

over Inference Time (0.099). For Effort, Required Skill Level received the highest weight (0.548), while Implementation Time (0.215) and Training Time (0.236) were viewed as equally relevant. Within Transparency, Visual Explanation (0.759) was considered far more important than Structural Explanation (0.241). Taken together, the results emphasize a strong user preference for predictive quality and intuitive interpretability in AI-based DSS selection.

### 5.3. Ranking of AI-based DSS archetypes

To answer RQ2, we combined the AHP-derived decision weights from RQ1 with measured values for each AI system characteristic to assess overall user preferences for the three DSS archetypes: Direct RUL, Indirect RUL, and Similarity-based Matching. Table 3 summarizes the measured values of all characteristics across archetypes. These include both objective measures (e.g., RMSE, Inference Time) and subjective assessments (e.g., Visual Explanation ratings derived from user rankings).

Using these values, we computed a weighted score for each archetype by multiplying each characteristic's normalized measurement with its corresponding AHP weight and summing across all characteristics. The resulting preference scores are shown in Fig 5.

Direct RUL and Similarity-based Matching were nearly equally preferred, with scores of 0.36 and 0.34, respectively. Indirect RUL received a lower score of 0.30. The preference for Direct RUL is largely driven by its strong Performance (0.46 weighted contribution), particularly in Predictive Accuracy. In contrast, Similarity-based Matching scored higher on

**Table 3. Measurement Results for High-Stake Maintenance AI-based DSS.**

| Decision Factor | AI System Characteristic | Direct RUL | Indirect RUL | Similarity-based Matching |
|---|---|---|---|---|
| Performance | Predictive Accuracy (PHM08) | 270 | 775 | 409 |
| | Predictive Accuracy (RMSE) | 13.51 | 21.23 | 14.17 |
| | Inference Time | 1562 μsec | 6006 μsec | 5345 μsec |
| Effort | Implementation Time | 8 hrs | 15 hrs | 25 hrs |
| | Training Time | 313.9 sec | 44.9 sec | 57.2 sec |
| | Required Skill Level | 69 hrs | 60 hrs | 73 hrs |
| Transparency | Structural Explanation | 0 | 1 | 1 |
| | Visual Explanation | 0.1 | 0.52 | 0.38 |

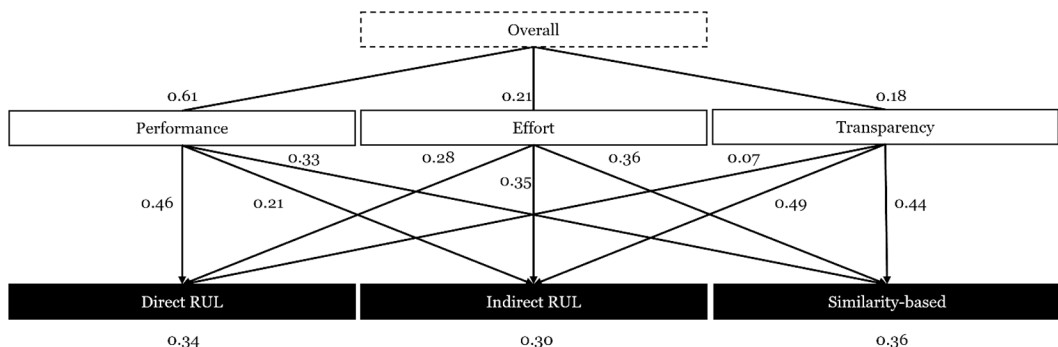

**Fig 5. Decision Weights for the Archetypes of AI-based DSS for Prognostics.**

Transparency (0.44 weighted contribution), especially for Visual Explanation. Despite the relatively high Transparency of Indirect RUL, its lower Performance and higher Effort diminished its overall appeal.

These results suggest that while Transparency and Effort are meaningful to users, Performance remains the dominant factor shaping DSS preferences in high-stakes maintenance contexts.

## 5.4. Impact of expertise on preferences

To address RQ3, we explored how users' domain expertise influences their preferences for different AI system characteristics and decision factors. Table 4 presents the breakdown of decision weights across low, medium, and high expertise groups.

Expertise had little impact on the overall preference for the three archetypes. All groups rated Direct RUL and Similarity-based Matching similarly and showed only slight variation in their evaluation of Indirect RUL. However, expertise did influence preferences for individual characteristics, particularly within the Transparency dimension.

Specifically, we observed that users with low and high expertise placed greater importance on Visual Explanation, whereas users with medium expertise were more receptive to Structural Explanation. This inverted-U relationship is further supported by the results of a probit regression (Table 5), which regressed the preference for Structural Explanation

**Table 4. Preferences for Decision Factors and AI System Characteristics Depending on User Characteristics.**

| Decision Factor | AI System Characteristic | All (*n*=120) | Low Expertise (*n*=47) | Medium Expertise (*n*=49) | High Expertise (*n*=24) |
|---|---|---|---|---|---|
| **Performance** | Predictive Accuracy | 0.901 | 0.933 | 0.841 | 0.923 |
| | Inference Time | 0.099 | 0.067 | 0.159 | 0.077 |
| **Effort** | Implementation Time | 0.214 | 0.209 | 0.230 | 0.201 |
| | Training Time | 0.236 | 0.245 | 0.242 | 0.211 |
| | Required Skill Level | 0.548 | 0.546 | 0.528 | 0.588 |
| **Transparency** | Structural Explanation | 0.241 | 0.234 | 0.315 | 0.139 |
| | Visual Explanation | 0.759 | 0.766 | 0.685 | 0.861 |
| **Overall (Decision Factors)** | Performance | 0.608 | 0.606 | 0.613 | 0.584 |
| | Effort | 0.210 | 0.238 | 0.176 | 0.236 |
| | Transparency | 0.182 | 0.156 | 0.211 | 0.180 |
| **Overall (Archetypes)** | Direct RUL | 0.362 | 0.365 | 0.360 | 0.357 |
| | Indirect RUL | 0.295 | 0.293 | 0.296 | 0.299 |
| | Similarity-based Matching | 0.343 | 0.342 | 0.344 | 0.343 |

**Table 5. Regression Results.**

| Variable | Dependent Variable: Structural Explanation (1) vs. Visual Explanation (0) |
|---|---|
| Intercept | −1.09 (0.99) |
| Industry | **−.80** (.27)** |
| Nr. of employees | .05 (.09) |
| Willingness to take risks | −.14 (.15) |
| Age | −.09 (.13) |
| Gender (female) | **.75** (.28)** |
| Expertise | **1.21* (.48)** |
| Expertise² | **−.23** (.09)** |
| LR Chi² | **20.52**** |
| Pseudo R² | .13 |

Positive coefficients indicate preference for Structural as opposed to Visual Explanation, standard error in parentheses, significant numbers in bold, *$p < 0.05$, **$p < .01$.

(coded as 1) versus Visual Explanation (coded as 0) on Expertise and control variables. The model reveals a significant quadratic effect of expertise (linear term: $\beta = 1.21$, $p < .05$; quadratic term: $\beta = -.23$, $p < .01$), as visualized in Fig 6.

These findings indicate that while novices and experts favor intuitive, instance-based explanations (Visual Explanation), users with moderate expertise may be more inclined to engage with the underlying system logic (Structural Explanation). No other significant effects of expertise on preference for Effort or Performance characteristics were observed. A power analysis using the GPower tool on the linear multiple regression for a fixed model with seven predictors shows that a sample size of $n = 89$ is sufficient for a medium effect size, which we exceed with our final sample size of $n = 120$. Hence, it is possible that weak effects between expertise and preferences for AI system characteristics other than Structural vs. Visual Explanation have not been detected by our analysis.

## 6. Discussion and implications

### 6.1. Discussion of decision factors and characteristics

The results of our study enable a critical reflection on the relevance of decision factors and AI system characteristics, decisions on prognostics archetypes for AI-based DSS, and a consideration of the influence of user characteristics on decision making regarding the relative importance of AI system characteristics.

**Importance of Decision Factors and AI System Characteristics.** As our overview in Table 4 shows, we found that *Performance* is the most important decision factor to address the focused selection problem regarding AI-based DSS to delegate decision-making tasks. Although this result was expected for the case of high-stake maintenance considering the theoretical grounding in transaction cost economics and resource-based view and related research (e.g., [7]), the magnitude of the decision weight was lower than expected in comparison to extant literature, which almost entirely suggests that the measures of Predictive Accuracy and Inference Time are the only AI system characteristics commonly used in evaluation [90].

As mentioned above, we found that *Effort* and *Transparency* are less but similarly important in comparison to each other. While this may sound surprising from a scientific point of view, it was to be expected from a theoretical as well as a practitioner's point of view. Effort introduces specific investments from an economic perspective that cannot be disregarded. Transparency relates to the issue of limited behavioral observability.

Only very few scientific articles are concerned with the actual cost of AI-based DSS implementations or the comparison of alternatives [91–93]. Required Skill Level accounted for more than 50% of the decision factor's weights highlighting

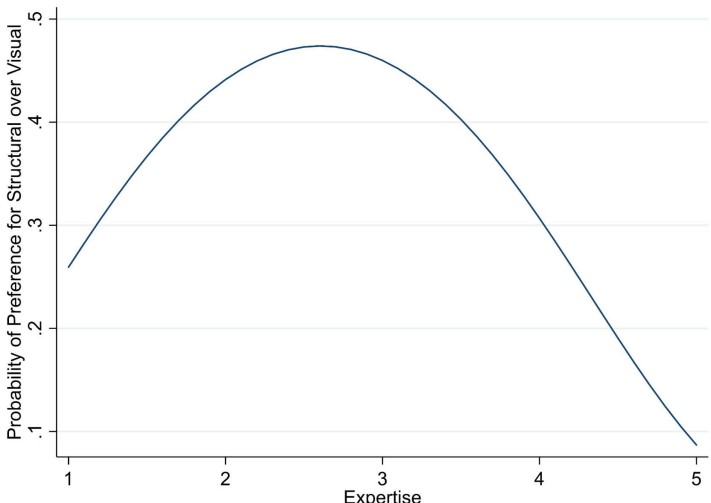

**Fig 6. Relationship between Expertise and the Preference for Structural over Visual Explanations.**

the difficulty of acquiring adequate talent in comparison to the rather mundane characteristics Implementation Time and Training Time. This focus on skill acquisition and labor cost can be partially explained by the fact that AI-based DSS typically introduce a novel degree of quality within a corporate information system landscape and due to its novelty, it is hard to capture all associated long-term benefits and cost savings. Nevertheless, people may attribute value to the investment subconsciously.

The interrelation of the decision factor Effort further complicates the choice of a specific AI-based DSS in two important ways. First, the amounts and outcomes of these efforts are uncertain. It is possible that the actual learning, implementation, and training efforts are higher than expected or that they yield a solution that does not meet performance expectations, possibly because the fit of an algorithm with a specific decision-making problem has been overestimated. Second, the efforts may be, to a large extent, specific for a given AI-based DSS, that is they are imperfectly transferable to another AI-based DSS. These two complications parallel the delegation problem in outsourcing settings. Outsourcing projects often require investment efforts in human knowledge (e.g., knowledge transfer at the project's start) [94] and in physical assets (e.g., integration systems) [95]. These efforts are uncertain and may transfer imperfectly to other sourcing settings (e.g., to an outsourcing arrangement with a different client or vendor) [96–98]. Given imperfect transfer, decision makers may be reluctant to make high amounts of these efforts in situations of high uncertainty to avoid sunk costs and a strong dependence on a vendor.

In a similar vein, AI-based DSS projects require investment efforts into human knowledge and into physical assets in the form of a trained AI-based DSS. While Required Skill Level drives the former effort, Implementation Time and Training Time drive the latter effort. It can be expected that decision makers want to avoid high amounts of these efforts especially when uncertainty is high. Uncertainty may arise from inscrutability of the algorithm and from lack of knowledge in the realm of the decision-making task. We discuss these two issues next. To objectivize as much as possible, in our research we only covered the investment cost of the outsourcing decision. In practice, this cost must be weighed against saving such as lower production cost that may differ from system to system and alter the weights towards the cost perspective. We assume this to be case in particular for low-stake cases, where a lack of Predictive Accuracy does not have the same consequences.

With regard to agency theory and the black-box architecture of contemporary AI-based DSS, Transparency becomes a major decision factor in assessing an agents work or outcomes. While earlier symbolic AI-based systems used hand-crafted and thus human-understandable rules, deep learning resembles the unknown of a new agent with prior qualifications but uncertain future performance. We refer to the growing body of research on AI explainability that investigates methods to foster acceptance of AI-inferred results by visualizing the AI models' inner workings (e.g., [99]). On this basis, we distinguish between two broad categories to visualize decision logic in our research taking into account user characteristics for information processing: the ability of a model to make its overall decision logic transparent (Structural Explanation) and the ability to justify individual predictions visually incorporating domain-specific knowledge already known to the user due to their experience (Visual Explanation). While there is a great variety to visualize explanations of complex models, we decided to differentiate Structural versus Visual explanations to investigate a key distinction to visually communicate the models' decision logic. Surprisingly at first, the Visual Explanation of the results seems to be much more important than the Structural Explanation. Yet, as we considered transparency from an end user's perspective and not from an engineer's perspective, subjects overall preferred Visual Explanation over Structural Explanation as it provides explanations that better correspond to the problem at hand and, thus, are typically preferred in production according to XAI research [5]. This is since the sheer number of sensor readings specific to our case can overwhelm even expert users, while a visualization gives a clear explanatory degradation trajectory that is understood by all maintenance experts.

**Decision on Archetypes AI-based DSS for Prognostics.** We depicted the results of the decision process in the lower part of Table 4 and show that across all attitude variations, *Direct RUL* and *Similarity-based Matching* are tied with scores of ~0.35 and *Indirect RUL* is considered inferior with a weight of 0.29. The choice weights of the overall decision factors that lead to those final scores are depicted in Fig 5 for all participants.

*Direct RUL* achieves its score purely by the performance advantage (0.46). This is in accordance with the fact that state-of-the-art black-box models based on deep learning achieve high task-based Predictive Accuracy. Improving the transparency of the model with XAI augmentations to justify the results and assign input variables weights towards the output would even further strengthen this approach. *Similarity-based Matching* only achieves mediocre performance but scores high in Transparency (0.44) and therefore is tied with Direct RUL at a normalized decision score of ~0.35. In comparison to the Direct RUL, which uses deep learning, Similarity-based Matching takes about three times as much Implementation Time than Direct RUL, however at a lower Required Skill Level. The low Implementation Time of the Direct RUL deep learning model can be attributed to the fact that deep learning has been gaining a lot of attention from non-AI professionals resulting in a considerable boost of high-level programming packages that allow building deep learning models more efficiently. In contrast, the enormous Training Time of the Direct RUL approach and the rather advanced Required Skill Level explains its low overall score in Effort (0.28) and, thus, its inferiority to the other approaches in this regard.

Although the *Indirect RUL* scores highest in Transparency, the lower importance of this decision factor as well as its low performance and the rather high effort renders it inferior to the other two archetypes.

Our results clearly show that the often-proposed trade-off between performance and transparency (e.g., 5), while existent, is heavily skewed towards performance in the case of high-stake decisions. Nevertheless, in those cases where explainability matters or even is a requisite, *Similarity-based Matching* provides an able alternative to *Direct RUL*.

Using the decision weights from RQ1, we can further explore trade-off relations of the decision factors for the hypothetical archetypes and create what-if scenarios. Fig 7 depicts the trade-off between Performance and Transparency. By using 0.05 increments of Performance (black bars) and 0.05 decrements of Transparency (grey bars) we can generate 21 hypothetical archetypes representing the full spectrum of implementation options. For example, archetype scenario #8 is created by the combination of Performance = 0.35 and Transparency = 0.65). By applying the AHP decision weights for both factors, trade-off lines for Performance (orange) vs. Transparency (light blue) can be constructed. This shows that only a small number of hypothetical combinations would be favored due to their Transparency (area A) in comparison to a

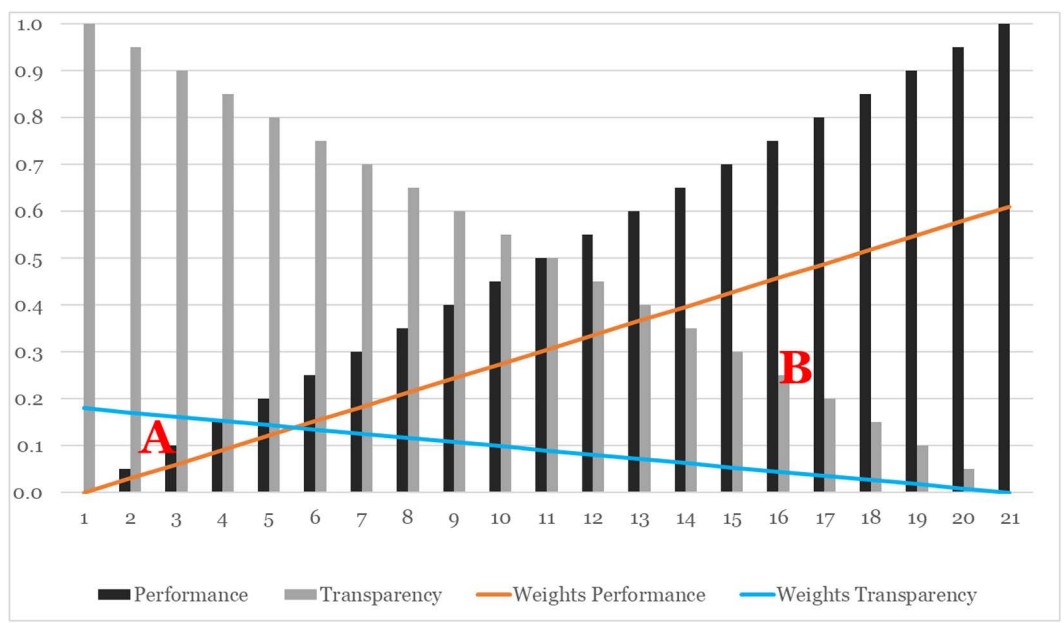

**Fig 7. Simulated Trade-off of Performance vs. Transparency.**

large number of Performance-dominated combinations (area B). The break-even point is skewed to the left side of the plot (between archetypes #5 and #6), while the break-even for equally distributed weights would be at archetype #11. Figures for the other two trade-offs involving Effort can be found in S1 Appendix Part C. While there were some minor differences in the responses of participants with different user characteristics, the coefficients of the regression analyses highlighted a significant relationship only between expertise and the preference for Visual versus Structural Explanations. That is, we observe a strong preference for Visual Explanations in users with low and high expertise, whereas users with moderate expertise demonstrate a lesser preference.

**Influence of Expertise on the Relative Importance of AI System Characteristics.** While surprising at first, the relationship can be explained by the differing needs of novices, intermediates, and experts. Novices seek guidance and use AI-based DSS to acquire basic knowledge and receive training and feedback. AI-based DSS enable them to make decisions normally associated with a higher level of expertise. AI-based DSS can give them step-by-step guidance and clear instructions to help them understand and navigate new tasks or situations that they cannot master yet due to their limited knowledge and experience in the domain [100]. They cannot yet ascertain the decision-making process but focus on the problem at hand as explained by a Visual Explanation. Similarly, but from a different angle, expert users do not necessitate extensive insights into the decision-making process. It is implicitly or explicitly clear to them and most prediction will correspond to their own expectations as they are used to act with high autonomy due to their extensive experience and a deep understanding of the nuances of the field [101]. A Visual Explanation is all they need to verify their judgements. Intermediates on the other hand require access to more advanced information to deepen their understanding and develop advanced problem-solving skills. They may need assistance in weighing options, evaluating alternatives, and assessing risks, as they possess some experience but may still struggle with more complex or ambiguous situations [81]. Hence, they want to gather as much information about the problem space as possible and, thus, value the more complex but more generalizing insight into the decision-making process provided by Structural Explanations over instances of explanations provided by Visual Explanations.

## Theoretical and practical implications

From our results and discussion, we can compile several implications for theory and practice. We discuss them in relation to the three research questions on decision factors, prognostics archetypes, and user characteristics.

**AI-based DSS selection can be framed as part of a delegation problem.** Our in-depth analysis reveals that the decision factors and AI system characteristics we derived from the extant theories of agency theory, transaction cost economics, and resource-based view can be applied for AI-based DSS selection and provide meaningful insights into the variables at work. In confirmation of extant literature, the results highlight that performance should never be the single factor in AI-based DSS selection. Transparency and Effort constitute relevant and practical additional variables in the selection process. Going beyond existing work, we defined a rigorous process to ascertain decision weights for these factors and we were able to attach tangible decision weights to all three decision factors Performance, Effort, and Transparency for the case of high-stake prognostics. This allows researchers as well as practitioners to develop their own scales for different use cases or their specific implementations.

**Performance can offset transparency and vice versa.** Our analysis of prognostics archetypes revealed that Direct RUL and Similarity-based Matching were on par regarding their overall value to participants. The analysis further revealed that this is due to the superior performance of Direct RUL and the predominant Transparency of Similarity-based Matching. Hence, at least for certain applications those factors can offset each other. In this respect, it is important to understand that Performance and Transparency are factors of the system and can be offset by superior user performance when revising the system decision due to extensive explanations. This has relevance for theory and practice when setting up studies and systems as well as legislation to mandate appropriate levels of observability when necessary. Simulations can create further insights into the trade-off decision as it is not symmetrical but skewed towards Performance. Effort revealed to be a factor that is comparatively uniform for all considered systems. While it is generally conceivable that it could influence trade-off decisions similarly in cases of novel technologies or skill deficits, we assume it to be rather an aggravating or facilitating factor than a decisive factor.

**Calibrate for user performance not system performance.** Our analysis of the human-information processing perspective implies AI-based DSS should be calibrated towards their users. While our research only highlighted the importance of providing a local over a global explanation for novices and experts and vice versa for intermediates, the implications are wider and not only tied to expertise. Extant research shows that there is better performance for trust-calibrated AI-based DSS by reducing uncertainty [102]. Moreover, there is evidence that AI-based DSS can be calibrated to improve user performance by decalibrating them and, thus, nudging users to scrutinize their decision (process) [103]. This opens manifold opportunities for AI-based DSS that need to be explored to optimize for user performance not system performance as the former is the measure that is relevant in real-world practice.

## 7. Conclusion

In this paper, we simulated a decision process and applied an AHP-based approach to investigate how users select AI-based DSS for high-stakes prognostic tasks, focusing on systems using black-box AI models. First, we framed the selection task as a delegation problem, drawing on agency theory, transaction cost economics, and the resource-based view to derive relevant decision factors. Second, we operationalized these factors in a measurement model. Third, we implemented three AI-based DSS archetypes in a high-stakes use case—predictive maintenance of aircraft turbines. Fourth, we conducted a user study to elicit decision weights and explore how user expertise influences preferences.

Our findings offer several key insights. Performance emerged as the most influential decision factor, with Effort and Transparency close behind. Among system archetypes, Direct RUL scoring and Similarity-based Matching performed best—each excelling in different attributes: Predictive Accuracy and Visual Explanation, respectively. Notably, user expertise moderated the importance of transparency strategies. Novices and experts favored different types of explanations, while intermediate users placed less emphasis on transparency overall.

Our study makes four contributions. First, we introduce a delegation-based perspective on AI-based DSS selection, offering a theoretically grounded lens to build on prior work in information systems outsourcing and decision support. Second, we provide empirically validated decision factors and operationalized measures that can inform future research and real-world decision processes. Third, our implementation and evaluation of AI-based DSS archetypes yield actionable insights into trade-offs between performance, effort, and transparency. Fourth, we identify the moderating role of user expertise, highlighting that transparency needs vary significantly across user types.

As with any research, our study faces limitations. First, while we designed the scenario to reflect high-stakes decision-making, we cannot fully replicate the psychological and contextual pressures of real-world settings. Second, we focused on end users as proxies for diverse organizational stakeholders; future research could explicitly incorporate perspectives from developers or decision-makers at other levels. Third, to maintain focus, we excluded broader information system adoption factors such as prior experience, hedonic motivation, or facilitating conditions. Fourth, we kept contextual variables such as environmental constraints constant to minimize bias, which limited our ability to analyze how these affect trade-offs. Fifth, while our performance measures are comparable within the case, they depend on specific technical configurations (e.g., hardware, software). Our results are therefore context-specific, though methodologically transferable. Sixth, preferences for structural and visual explanations may be shaped by users' familiarity with specific visualization techniques, which we did not assess directly. Nonetheless, our use of industrial maintenance professionals helps ensure realism and lays a foundation for future studies on the impact of individual differences.

While we applied the decision process to a specific case to demonstrate it, we think it can be used for the general problem of choosing an AI-based DSS for a case with high-stake decisions. That is, the results of this study can be used in both normative and prescriptive decision analysis regarding AI systems as a starting point to investigate the trade-offs of decision factors for specific domains or even domain-independent contexts. In addition, our study also raises awareness of the preferences of practitioners when it comes to selecting an AI system.

## Supporting information

**S1 Appendix.  Appendix.**
(PDF)

**S2 Dataset.  Dataset.**
(XLSX)

## Author contributions

**Conceptualization:** Jonas Wanner.

**Data curation:** Jonas Wanner.

**Formal analysis:** Kai Heinrich, Oliver Krancher, Jonas Wanner.

**Methodology:** Jonas Wanner, Patrick Zschech.

**Project administration:** Christian Janiesch, Philip Stahmann, Jonas Wanner.

**Supervision:** Christian Janiesch.

**Writing – original draft:** Kai Heinrich, Christian Janiesch, Oliver Krancher, Philip Stahmann, Jonas Wanner, Patrick Zschech.

**Writing – review & editing:** Kai Heinrich, Christian Janiesch, Oliver Krancher, Philip Stahmann, Patrick Zschech.

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
