## [Decision Letter · Decision Letter 0]

Dear Dr. Janiesch,

Thank you for submitting your manuscript to PLOS ONE. After careful consideration, we feel that it has merit but does not fully meet PLOS ONE’s publication criteria as it currently stands. Therefore, we invite you to submit a revised version of the manuscript that addresses the points raised during the review process.

Please assure that all the comments pointed-out by the reviewers would be addressed in the revisited manuscript.

We look forward to receiving your revised manuscript.

Kind regards,

Alessio Luschi, Ph.D.

Academic Editor

PLOS ONE

3. We are unable to open your Figure file [image3.eps and image6.eps]. Please kindly revise as necessary and re-upload.

Reviewers' comments:

Reviewer's Responses to Questions

**Comments to the Author**

1. Is the manuscript technically sound, and do the data support the conclusions?

Reviewer #1: Yes

Reviewer #2: Yes

2. Has the statistical analysis been performed appropriately and rigorously?

Reviewer #1: Yes

Reviewer #2: I Don't Know

3. Have the authors made all data underlying the findings in their manuscript fully available?

Reviewer #1: Yes

Reviewer #2: Yes

4. Is the manuscript presented in an intelligible fashion and written in standard English?

Reviewer #1: Yes

Reviewer #2: Yes

Reviewer #1: The manuscript presents an interesting study, with a clear and well-developed writing style that effectively guides the reader through the research process and researchers' intentions. However, I believe that the manuscript would benefit from significant revision to improve its readability and conciseness.

First of all, the manuscript is at times overly detailed, making it dense and repetitive. For example, the study’s objectives are restated multiple times throughout the text. These repetitions in the whole text should be reduced for a more streamlined narrative.

Moreover, certain sections, such as those describing the research process in the conclusions, should be made more concise, focusing on critical insights rather than reiterating procedural details.

The “Study Design” and “Results” sections should be clearly separated. Currently, these are merged, making it difficult to distinguish the methodological framework from the findings.

The “Study Design” section should include the experimental procedure, measures, and sample characteristics in a structured manner.

The inclusion of footnotes is not permitted according to the journal’s guidelines and should be removed.

For clarity the paragraph 3.2.4 should be anticipated before table 1 so that the table is at the end of the chapter as a summary of the dimensions.

Overall, the manuscript presents an innovative study that contributes valuable insights to the field of AI and Decision-making. However, addressing the issues related to structure, and conciseness would significantly enhance its readability and impact. I recommend a thorough revision to refine the manuscript accordingly.

Best regards

Reviewer #2: This paper addresses the challenge of selecting an appropriate Decision Support System for a given organisation. The authors model this as a delegation problem, identify relevant system features and organise them using the Analytical Hierarchy Process method. In a user study, the features are weighted and then matched against different system types to see which one(s) would fit the users top desiderata best. In a sideline, the authors assessed users’ preference for different XAI modalities in relation to their level of expertise.

This paper presents an interdisciplinary approach of tackling a real-world problem and it can serve as a blueprint for many related issues of matching AI systems to organisations’ needs although some of the details of this particular case seem weak.

It is notoriously difficult to do justice to interdisciplinary contributions in a review. As an AI expert, I can say very little about the setup and validity of the user study. But overall, the quality of writing is very high and the argumentation seems solid. Therefore, I judge this contribution as publishable.

Overall, I have the impression that the paper is a little bit overselling, oftentimes stating observations and assumptions as irrefutable although many different alternatives come to mind that would also be valid (e.g., the selection of the characteristics). In this direction a few observations from the AI foundations in chapter 2.1:

p. 7, l. 16 “AI-based DSS use analytical models that are trained by a learning algorithm based on training data.” for the sake of completeness, one should add that “Today, the most common form of DSS uses analytical …”, there is a whole history of others DSS such as knowledge-based or hybrid systems.

p. 8. l. 18 “automated feature learning” This could be misinterpreted as some readers might expect that ANNs would automatically output human-readable features. Maybe it would be better to use the term “pattern recognition” here if this is what is meant.

p. 9, l. 9 “algorithmic decision logic”. There are many different ways of how XAI can support DSS. As above, “algorithmic decision logic” might raise the expectation in some readers that one can, e. g., easily derive a decision tree from an ANN-based DSS system. Maybe one could say that XAI has the goal to enable humans to better understand how the system arrives at its outputs.

More general comments (some of the criticism you already address in the discussion):

When you formulate DSS selection as a delegation problem, you mention a machine maintainer and their specific needs such as robustness of the predictions or information about spare parts that might be needed for a mainiantance. You also mention principals, decisions makers and other involved persons. For all these stakeholders, the features performance, effort, transparence and others might mean something different and have different relevance. The machine maintainer might not worry about training time of a model as long as he can work with a suitable model when he needs its. Transparency for a management and scheduling person is needed on another level as compared to a repair person.

Instead of discussing the decision factors in the void, wouldn’t it have made sense to define certain personas like the ones mentioned above and indicate for each factor for which persona it is interpreted?

Likewise, some factors can only be weighed when the properties of the scenario at hand are known. If, e.g. a turbine is changing over time, it might be needed to re-train a model every three weeks as compared to a situation where a turbine stays the same for several years. In the first scenario, training time might be much more critical than in the latter.

Finally, the binary option visual vs. structural explanation seems a bit oversimplified. You introduce global and local decisions and then turn to the specific terms structural vs. visual. Why are you mixing categories? As above, it would probably be much easier to discuss the pros and cons given personas and some specific cases.

Summing up, as a blueprint, the approach has its charme and qualifies for publication, the detailed choice of features, weights and also the mapping to the models is not very convincing. This could be presented much more as *one exemplification* of how such a decision matrix could be filled and decision executed.

**Do you want your identity to be public for this peer review?** For information about this choice, including consent withdrawal, please see our Privacy Policy

Reviewer #1: No

Reviewer #2: No

---

## [Author Response · Author response to Decision Letter 1]

13 Jun 2025

Dear Sir or Madam,

we have revised the manuscript according to the review letter. Please review the cover letter and reply to reviews.

Regards

the authors

---

## [Decision Letter · Decision Letter 1]

Decision Factors for the Selection of AI-based Decision Support Systems - The Case of Task Delegation in Prognostics

PONE-D-25-00204R1

Dear Dr. Janiesch,

We’re pleased to inform you that your manuscript has been judged scientifically suitable for publication and will be formally accepted for publication once it meets all outstanding technical requirements.

Kind regards,

Alessio Luschi, Ph.D.

Academic Editor

PLOS ONE

Additional Editor Comments (optional):

Reviewers' comments:

Reviewer's Responses to Questions

**Comments to the Author**

Reviewer #2: All comments have been addressed

Reviewer #3: All comments have been addressed

2. Is the manuscript technically sound, and do the data support the conclusions?

Reviewer #2: Yes

Reviewer #3: Yes

3. Has the statistical analysis been performed appropriately and rigorously?

Reviewer #2: I Don't Know

Reviewer #3: Yes

4. Have the authors made all data underlying the findings in their manuscript fully available?

Reviewer #2: Yes

Reviewer #3: Yes

5. Is the manuscript presented in an intelligible fashion and written in standard English?

Reviewer #2: Yes

Reviewer #3: Yes

Reviewer #2: As in my original review, I consider this article ready for publication.

Remaining typos:

p11, l16/17: "require" => "requires", missing space before "Transparceny"?

p29, l19: missing space before "A".

Reviewer #3: I consider that the authors have collected the main suggestions. As a result of these efforts, this paper is now sufficiently suitable for publication. Thus, I recommend accepting it in its current form.

**Do you want your identity to be public for this peer review?** For information about this choice, including consent withdrawal, please see our Privacy Policy

Reviewer #2: No

Reviewer #3: No

---

## [Editor Report · Acceptance letter]

PONE-D-25-00204R1

PLOS ONE

Dear Dr. Janiesch,

I'm pleased to inform you that your manuscript has been deemed suitable for publication in PLOS ONE. Congratulations! Your manuscript is now being handed over to our production team.

Kind regards,

on behalf of

Dr Alessio Luschi

Academic Editor

PLOS ONE